# A Comparative Analysis of Risky Sexual Behaviors, Self-Reported Sexually Transmitted Infections, Knowledge of Symptoms and Partner Notification Practices among Male and Female University Students in Pretoria, South Africa

**DOI:** 10.3390/ijerph18115660

**Published:** 2021-05-25

**Authors:** Mathildah Mpata Mokgatle, Sphiwe Madiba, Lindiwe Cele

**Affiliations:** 1Department of Biostatistics, School of Public Health, Sefako Makgatho Health Sciences University, Ga-Rankuwa 0208, South Africa; lindiwe.cele@smu.ac.za; 2Department of Environmental and Occupational Health, School of Public Health, Sefako Makgatho Health Sciences University, Ga-Rankuwa 0208, South Africa; sphiwe.madiba@smu.ac.za

**Keywords:** risky sexual behaviours, knowledge of STI/HIV, university students, risk perception, partner notification, South Africa

## Abstract

The surge of sexually transmitted infections (STIs) among young people is of public health importance, and the notification and treatment of sex partners after the diagnosis of an STI is a public health approach to prevent and reduce further transmissions. There are limited studies that investigate partner notification among young people in general, and university students in South Africa in particular. We investigated self-reported STIs and partner notification practice, intentions, and preferences among university students. We also assessed their STI knowledge and risky sexual behaviour in relation to STIs. The study was a descriptive cross-sectional survey that used multistage sampling to select 918 students across the five schools of a health sciences university in South Africa. Descriptive statistics and bivariate logistic analysis were performed using Stata IC version 16. More males (54.1%) than females were currently in a sexual relationship (47.3%), more males reported multiple sexual partners (*n* = 114, 46%), engaged in transactional sex (*n* = 13, 5.3%), and had one-night stands (*n* = 68, 28.1%) in the past 12 months (*p* = 0.001). Moreover, half (55.9%) had poor knowledge of STIs with an overall mean knowledge score of 2.9 ± 2.0, and the majority (85.8%) perceived themselves to be at low risk of acquiring STIs. The odds of intentions to disclose an STI infection to a sexual partner and delivering a partner notification slip to ex-sexual partners were not statistically significant (*p* = 0.95; *p* = 0.10), with the likelihood of disclosure being 1.3 times for female students compared to males. Female students were 1.5 times as likely to prefer a doctor to send an SMS notification to their sexual partners (*p* = 0.02) compared to their male counterparts, while the preference of an SMS notification was 41% (*p* = 0.03) among female students. Students engaged in risky behaviours but had a low perception of the risks of acquiring STIs. Although they had preferences of different methods of partner notification, both male and female students preferred SMS partner notifications from a doctor, even though women were in the majority. Health care providers should put in place interventions so that young people can safely inform their partners about STIs.

## 1. Introduction

The World Health Organization (WHO) estimates that globally more than one million cases of sexually transmitted infections (STIs) are contracted daily. Some of these infections are asymptomatic and represent a risk of infection to sexual partners [1]. Earlier WHO estimates indicated that, globally, one-third of over 340 million new STI cases occur in people under the age of 25 years [2]. In South Africa, the burden of disease due to STIs is one of the largest in the world, with an estimated 23,175 million new cases of syphilis, 4.2 million cases of chlamydia, and 6.2 million cases of gonorrhoea reported in 2017. Similar to global trends, the new cases of STIs were reported to have occurred mainly among the 15- to 49-year-old individuals. The high STI prevalence among young people observed globally highlights the importance of global interventions to improve sexual and reproductive health in this population [3].

In developing countries, young people are at risk of STIs due to their risky sex behaviours, limited preventive practices [4], and lack of adequate knowledge of STIs [5]. Risky sexual behaviour in young people in the developing world is an important health, social, and demographic concern [6]. Most university students are in the youth age category and are categorised under the most at risk population group due to their inclination to be engaged in risky sexual behaviour, which leads to acquiring STIs [7,8,9]. They are at high risk of acquiring STIs and HIV due to an early age of sexual debut and inconsistent condom use [10,11]. The earlier age of sexual debut and low condom use among young people, including university students, has been associated with a reduction in the success of prevention interventions [12,13].

Other studies reported that university students are likely to engage in unprotected sex, have multiple sex partners, engage in sex with older partners, engage in casual sex, and have sex under the influence of alcohol or drugs [6,7,12,13,14]. In South Africa, 61% of students in the institutions of higher education are in a sexual relationship and 15% of them have sex partners who are older than themselves by five or more years [15]. Age-disparate relationships have been shown to increase the risk of HIV infection and other STIs [16,17]. Furthermore, only 55% of university students in South Africa reported condom use during their last sexual encounter with their regular partner, and 57% reported consistent condom use [15].

Studies noted the other factors that increase their susceptibility to STIs and HIV as a lack of knowledge of HIV and other STIs, and poor access to preventive services [11]. A study investigating the knowledge of STIs amongst students reported that 99% knew about HIV, but less than 50% knew about other STIs [18]. In another study, university students were concerned about pregnancy rather than STIs and HIV infection. Having a good knowledge of STIs is one of the factors that protects students from acquiring STIs [7]. It is therefore important to enhance the students’ knowledge and awareness towards STIs to prevent and control the spread of STIs [19].

According to the WHO, a significant proportion of STIs are asymptomatic globally [1]. Studies have noted a high burden of asymptomatic infections among young people in KwaZulu-Natal [3,20]. This is a cause for concern for the prevention of new infections in South Africa, since the asymptomatic nature of STIs make the control and treatment of them complicated. Despite the high and frequently asymptomatic STI infections among South African young people, the country’s STI treatment guidelines employ syndromic management. This is despite its limited ability to detect asymptomatic STI cases [21] and its reliance on individuals to report STI signs and symptoms. An important approach of syndromic management is patient-initiated partner notification using notification slips issued during the consultations with health care providers in primary health facilities. Patient-initiated partner notification (PN) is when the index cases inform their partners themselves [22,23], whereas provider-initiated partner notification is when a trained health professional contacts sexual partners for the index case using electronic messages, such as short message service (SMS) [24,25].

Research has noted that patient-initiated partner notification interventions have had barriers to their uptake, because index cases under-reported the number of sexual partners, due to fear of moral judgement [26,27]. Under-reporting also results from the limited number of slips issued by health care providers. The amount of time involved in counselling and educating the patients is a major limitation of the PN strategy [26]. The lack of strategies for health care providers to follow-up sex partners also compromises STI control [23]. Nevertheless, research conducted in South Africa shows that up to 50% of individuals diagnosed with STIs do not intend to notify their partner about the STI. A fear of negative partner reactions and violent responses are among the most common perceived barriers to partner notification [23]. As such, provider-initiated partner notification electronic messages have been adopted to expand the partner notification services [24,26].

There are limited studies that investigate partner notification among young people in general, and university students in South Africa in particular. In response to the dearth of data, we investigated self-reported STIs and partner notification practices, intentions, and preferences among university students in Pretoria. Furthermore, the knowledge, attitude and preventive practices of students on HIV has been extensively studied [28]; however, most of the literature does not address the knowledge and preventive practices of STIs. As such, we also assessed their STI knowledge and risky sexual behaviour in relation to STIs. The failure to inform their sex partners of their exposure to STIs increases the risk of STI transmission to other sexual partners who remain asymptomatic [29]. Since risky sexual behaviours are linked with an increased risk of HIV infection, understanding risky young people’s sexual behaviours is crucial to prevent and control the HIV epidemic.

## 2. Methods

### 2.1. Study Design and Population

The study setting for this cross-sectional design was a university in Pretoria, South Africa. The university has five schools (Medicine, Pharmacy, Oral Health, Health Care Sciences, and Science and Technology), and offers health sciences programmes. During the period of data collection, September 2017 to February 2018, the university had enrolled 7450 students. The university has one teaching referral hospital and the lead Clinical Management Centre of primary health care clinics in the neighbouring districts.

The study population consisted of undergraduate and postgraduate students enrolled at the university during the study period. With a population size of *n* = 7450, we treated the five schools within the university as a unit to ensure the representativeness of the data proportional to the number of students. The minimum sample was *n* = 181 students per school using Raosoft® (Online Sample Size Calculator; Raosoft inc., Seattle, WA, USA) for sample size calculation at 95% confidence level and 5% margin of error. The total estimated sample size in all the five schools was *n* = 905 students. A multistage sampling technique was used to select programmes and students. First, a sampling frame was developed from the list of programmes in each school. Programmes were selected from the school by simple random sampling and students at the various levels of study in each programme were selected by systematic random sampling.

### 2.2. Measures

The data were collected through a structured self-administered questionnaire. We collected information on sociodemographic, sexual behaviour and knowledge about STIs, ever diagnosed STIs, and partner notification awareness, experiences, and intentions.

Risky sexual behaviour was assessed with a 26-items measure including questions on condom use in the last sexual act, unprotected sex, number of sex partners, concurrent partnership, and transactional sex. Students were classified as having engaged in high-risk sexual behaviour if they practised at least one of the above-listed sexual behaviours. To assess the level of risk perception, a five-item three-scale Likert scale was employed. The students were asked how worried they were of getting HIV and the chance of contracting STIs. The response was categorised as 0 = not likely to contract the disease and was categorised as having low risk perception, and 1 = likely to contact the disease as having high risk perception.

A 14-item measure was used to assess STI knowledge and the questions were phrased as multiple choice to allow participants to select the correct response. Some knowledge questions asked whether a person infected with an STI could have no symptoms and what the common STIs symptoms are. Other knowledge questions were asked to give a response of “Yes” or “No” and the correct response was coded “1” and the incorrect or non-response was coded “0”. A total knowledge score was computed by summing the scores of each question. For each student, the possible total knowledge score ranged from 0 to 6. The students also reported on their experience, if any, of an STI diagnosis during the last 12 months and the common symptoms they experienced.

The partner notification questions were compiled in a five-item measure. The questions that covered intentions to notify sexual partners if they had an STI were assessed using seven items asking whether they would notify their partner if they had an STI and the preferred partner notification method. Responses were categorised as “Yes”, “No” and “Not sure”. For prior partner notification experiences, students were asked whether they had ever informed a sexual partner that they had been diagnosed or treated for an STI, and asked them whether a sexual partner had ever informed them of an STI.

### 2.3. Data Collection 

The data were collected using a self-administered standardised questionnaire in English. The questionnaires were administered by trained research assistants who distributed the questionnaires to students after a lecture in their lecture halls. The students completed the questionnaires individually but the research assistants were available to check for the completeness of the questionnaires. The questionnaire was developed by referring to previous tools and constructs obtained from the review of the literature on STIs, risky sexual behaviours, and partner notification [22,30,31].

One day of training was given for the research assistants on how to select the study students, on the objectives of the study, the content of the questionnaire, on confidentiality, and on administering informed consent. The questionnaire was pretested with 30 students (3.5%) of total expected sample and was amended based on the feedback received.

### 2.4. Data Analysis 

The analysis of the data was done using the STATA IC version 16.0 statistical package (STATA Corp., College, TX, USA). Descriptive statistics such as frequencies, percentages, and proportions were computed to describe the study variables. Socio-demographic, risky behavioural characteristics, and partner notification intentions were compared by sex. To determine students’ knowledge of STIs, responses to six selected knowledge questions were tabulated and reported as proportions, and the mean and standard deviations (SD) computed from the total knowledge score. The mean was used to categorise knowledge levels. Knowledge scores above the mean were categorised as good knowledge, while scores below the mean were categorised as poor knowledge.

The variable sex (male or female) was the dependant variable for the study, and the variables within the constructs of socio-demographics, sexual behaviour, knowledge of STIs and partner notification were the independent variables. The Pearson Chi-square was used to examine the difference of demographic and behavioural characteristics between male and female participants. The association between variables was measured by using bivariate logistic regression. A significance level of 5% was accepted. The strength of association between dependent and independent variables was described using odds ratio at 95% confidence interval (CI). Statistical significance was set at *p* < 0.05 for all variables. 

### 2.5. Ethical Considerations 

The Research Ethics Committee of Sefako Makgatho Health Sciences University (SMUREC/H/256/2017) reviewed the protocol and gave an ethical clearance. The relevant university authorities granted permission to conduct the study. Informed consent was obtained from all students Participation was voluntary, including the right to withdraw from the study without any preconditions. For anonymity, no identifying information was collected and the data file was password protected, with access limited to the lead investigator.

## 3. Results 

### 3.1. Characteristics of Students 

A total of 918 students completed the questionnaire. Most (643/70.3%) of the students were female, their mean age was 21 years, SD = 2.3, and almost half (442/48.6%) of them were between the ages of 17 and 20 years. Only one quarter (227/25.2) of the students were in the first year of study, 455/49.7% were residing on campus, almost all of them were undergraduates (94.5%), and 91.3% were from health sciences schools. Half (459/50.5%) of the students were sexually active in the 12 months prior to the survey. Of the sexually active students, 172/28.6% had multiple sex partners in the past 12 months, and 192/31.7% had concurrent sexual partners. Most (424/78.5) had been in a relationship for less than five years, with over a third (189/35%) of them being in a relationship for less than a year (Table 1).

### 3.2. Risky Sexual Behaviour 

With respect to risky sexual behaviour, more males (54.1%) than females were currently in a sexual relationship (47.3%), more males reported multiple sexual partners (*n* = 114, 46%), engaged in transactional sex (*n* = 13, 5.3%), and had one-night stands (*n* = 68, 28.1%) in the past 12 months. The differences were statistically significant (*p* < 0.001). Of the sexually active students, 269 (39.3%) reported not using a condom, and more (44%) female students compared to 30.3% of males did not use a condom (*p* < 0.001).

Compared to the female students, more males were comfortable to purchase condoms (80.6% vs. 63.7%) and to get condoms from a public place (65.8% vs. 48.1%) without feeling embarrassed (Table 2). About a quarter (189/24.3%) always carry a condom, and more males always carry a condom with them (40.9% vs. 16.3%). These differences were statistically significant (*p* < 0.001). Overall, the majority (718/93.4%) of both male and female students were confident to suggest condom use to partner (92.6% vs. to 93.7%).

Slightly more than half (497/56.1%) had tested for HIV in the last 12 months, and more females had tested for HIV (59.8% vs. to 47.0%) and knew their partner’s HIV status (60.7% vs. 48.4%). The differences were statistically significant (*p* < 0.001). Overall, compared to the female students, the males reported significantly higher rates of risky sexual behaviours. 

### 3.3. Perception of Risk and Preventive Practices on STIs

On the perception of the risk of contracting STIs and HIV, the results showed that a third (265/29.8%) of the students perceived their risk of contracting HIV as low. There was a statistically significant difference with regard to the perception of the risk of HIV. Relative to their male counterparts (23.1%), more females (32.5%) reported a low risk of contracting HIV (*p* = 0.012). The majority (745/85.8%) reported a low risk of contracting STIs, and the difference was not statistically significant. A high proportion were likely to discuss HIV testing with their partner (75.1%), to refuse sex if the partner did not want to use a condom (73.1%) and 71.4% would ask their partner to go for HIV testing with them. Compared to the male students, the females were more likely to refuse condomless sex (79.7% vs. 61%), more likely to discuss HIV testing (78.5% vs. 67.2%), and more likely to ask their partner to test together (76.8% vs. 67.2%). The differences were statistically significant. The majority (587/72.5%) of the students were of the view that it is very important to tell their partner about an STI diagnosis so that they could seek treatment (Table 3).

### 3.4. Knowledge of Selected STI Symptoms 

The students’ knowledge of the selected six STI facts is summarised in Table 4. Overall, the students’ knowledge of STI symptoms was low (55.9%), with an overall mean knowledge score of 2.9 out of 6. Relative to the males, more females knew that pain during intercourse is an STI symptom (35.6% compared to 27.9%), and the difference was statistically significant (*p* = 0.024). There was a marginally significant difference in the proportion of females compared to males who reported knowledge of genital ulcers or open sores as an STI symptom (45.1% and 38.2%, respectively) (*p* = 0.055). There were no statistically significant differences in the proportions of males and females with knowledge of the following symptoms: itching in the genital area (*p* = 0.183), discharge (*p* = 0.301), and pain while passing out urine (*p* = 0.459). Nevertheless, a high proportion of both male (77%) and female (70.6%) students correctly indicated that STIs can be asymptomatic (*p* = 0.054).

### 3.5. Self-Reported STIs Symptoms and Partner Notification Practices 

Only 4.68% reported having STI symptoms in the 12 months preceding the survey (Table 5). The most common STI symptoms were itching around the genital area (9/34.6%), discharge (7/26.9%), pain while passing urine (6/23.1%), and genital ulcers or open sores (2/7.7%). There was no difference in reporting having STIs symptoms between the female and male students. Most (23/62.2) informed a partner of an STI, but less than a third (7/28%) had been informed of an STI by a partner. The majority consulted for the diagnosis and treatment of the STI symptoms, and 24/33.3% received a PN slip from the health provider. Of those who did not deliver a notification slip to their partners, more (13/92.8%) expected negative outcomes if they were to inform their partners.

### 3.6. Partner Notification Intentions 

Concerning knowledge about STI partner notifications, slightly less than a quarter (186/21.6%) knew of STI partner notifications, 141/23.5% of the female students and 44/17% of the males knew of STI partner notifications (*p* = 0.035). 

Overall, the majority (697/85.2%) of the students would notify their partners if they were infected with an STI, 799/95.4% would deliver a PN slip, 376/45.4% would notify their ex-partner, 549/66.2% would find it easy to receive a PN slip that requests them for treatment, and 310/37.4% would find it easy to deliver the PN slip to a partner. Two thirds (488/59.4%) of the students preferred that an SMS from a doctor be sent to their partner, inviting them to come for STI treatment, and 442/53.4% felt that an SMS would work better to notify a partner to come for STI treatment.

Bivariate logistic regression at a 95% confidence interval showed that the odds of intentions to disclose an STI infection to a sexual partner and the odds of delivering a PN slip to ex-sexual partners were 1.38 times more among the female students compared to their male counterparts (CI = 0.95–2.01). The female students had a 57% probability of delivering PN slips to their sexual partners compared to their male counterparts, at a confidence interval of 0.23 to 0.84. The data showed a low probability of 46% of ease of delivering a PN slip, as both the female and male students reported that it not easy to deliver partner notification slips to sexual partners (CI = 0.47–0.92). Female students were 1.5 times more likely to prefer a doctor to send an SMS notification to their sexual partners (CI = 0.23–0.41) compared to their male counterparts, while the probability of accepting the use of an SMS notification was at a low of 41% (0.23–0.41) for both the male female students (Table 6).

Over half (435/52.7%) of the students preferred their partner to notify them face-to-face if they were diagnosed with an STI, 262/31.7% preferred to receive an SMS from the clinic, and 129/15.6% preferred to receive a notification slip from their partner (Figure 1).

## 4. Discussion

This study reports on self-reported STI symptoms, and partner notification practices and intentions among health sciences university students in South Africa. The study found a low overall prevalence of self-reported STI symptoms of 4.6% among both the male and female students. In agreement with previous studies [32], the low prevalence of STIs reported might not be a true reflection of the magnitude of STIs since the data were self-reported by the students. The researchers suggest that young people may not consider their symptoms important, through a lack of awareness, which might result in the low prevalence of STIs [33], while a lack of familiarity with STI symptoms might exacerbated the under-reporting of symptoms [20]. The data showed that more males compared to females were in a current sexual relationship, had multiple sexual partners, engaged in transactional sex, and had one-night stands. The evidence of risky sexual behaviours of males versus females has been cited in the literature over time. 

We further found that knowledge about selected STI symptoms was minimal (score 2.9) among both the male and female students, despite the fact that a large number of them had heard about STIs and the majority were studying medicine, nursing, dental and oral health, and allied health. Moreover, 42% were senior students in their third year of study and above. Whilst only less than half of the students knew selected STI symptoms, three quarters (71%) knew that STIs could be asymptomatic. Our finding is consistent with multiple other studies that found varying levels of knowledge of STIs among university students and young people [34,35,36,37]. However, other studies reported low levels of students who knew that STIs could be asymptomatic [32,38]. Not knowing that STIs can be asymptomatic explains the issue of missed opportunities in early infection, high STI transmission to sexual partners, and high prevalence of STIs in the country [39]. 

Overall, half (50.5%) of the students had sex in the last 12 months and the results indicated that about 28.6% had multiple sex partners, whereas 39% of them practiced unsafe sex. The prevalence of multiple sex partners recorded in this study is low when compared with that of previous studies conducted elsewhere [12,19,22,32,37,40]. Consistent with other studies, the risky sexual behaviour was significantly higher in the male students compared to the females [41]. Other studies reported contrasting findings. Afriyie and Essilfie [42] found that the odds of engaging in risky sexual behaviour were higher for both males and females. In agreement with a number of previous studies [12,22,32,37], the use of condoms in this study was low for both the male and female students, but significantly more female students reported unprotected sex. The prevalence of unprotected sex practice was lower compared with findings that have been reported in other studies [12,40].

Despite the high risky sexual behaviour observed in the current study, the majority (85.8%) of the students perceived themselves to be at low risk of STIs, and a third (29.8%) perceived their risk of contracting HIV as low. The difference in the low risk perception of contracting STIs despite high risk behaviour was statistically significant, with more female students reporting low risk perception compared to males. Similar behaviours among young people were reported elsewhere [37,43]. Research has noted that young people may underestimate their risk of contracting STIs because of their low level of knowledge about STIs and what should be considered as risk factors for STIs, which results in the failure to link risky sexual behaviour to the likelihood of contracting STIs [37,43,44] Health promotion programmes should provide students with more information about STIs to increase young peoples’ ability to accurately measure their risks relative to their sexual behaviour [37,43]. However, most of the students had acceptable preventive practices for STIs, and almost half (49.5%) had had no sexual intercourse in the 12 months prior to the study. We found that a significant proportion (73.1%) would refuse condomless sex, and significantly more females would refuse condomless sex (79.7% vs. 61%) and 93.4% were confident to suggest condom use. Of concern is that only 24.4% carry a condom with them [19]. It was also found that 63.8% of the students do not keep a condom in their pocket to protect themselves, despite the relative high practice of casual sex.

This study showed that a small proportion (21%) of students have heard about partner notification, but a significantly high proportion (94.9%) of them knew the importance of notifying a sexual partner once diagnosed with an STI. Concerning partner notification intentions, it is noteworthy that the majority (85.2%) of the students reported that they would notify their partner if they themselves were diagnosed with STIs and 95.5% would deliver a notification slip to their partner. However, when asked how easy that would be, we found that two-thirds (66.2%) of the students would find it easy to deliver a notification slip to a partner and more than half (59.4%) of them preferred that an SMS from a doctor be sent to their partner notifying them about an STI. The preference for sending an SMS from the doctor to a partner instead of a slip may be an attempt to evade a partner’s adverse reaction, or it could be an indication of fear of being rejected or judged [22]. Mokgatle and Madiba [39] reported similar observations of conflicting views among adult men regarding informing partners should they be diagnosed with an STI, preferring the delivery of a notification slip to the partner. 

While the students were not inclined to notify a partner about an STI themselves if they were diagnosed with STIs, slightly over half (52.7%) preferred their partner to notify them themselves if they were diagnosed with an STI. The finding is consistent with reported preferences for partner notification among adults [39] and adolescents in STI clinicsh [45]. Where an SMS is concerned, only a third (31.7%) preferred to receive an SMS from the clinic notifying them about a partner’s STI diagnosis. A small proportion (15.6%) preferred to receive a notification slip from their partner. According to Mokgatle and Madiba [39], the preference of both direct partner notification and SMS from the clinic is an indication that these methods are necessary for STI control and STI notification. A systematic review investigating the acceptability and efficacy of partner notifications found that partners are more likely to seek STI treatment when notified by direct patient referral [46].

Of those who were diagnosed with STIs in the 12 months preceding the survey, 62.2% informed their partner of the diagnosis and a third (33.3%) received a notification slip from the health provider during the consultation. Low issuing of partner notifications is consistent with research that one of the limitations of the partner notification strategy is the limited number of notification slips issued by health care providers [25,26]. We found that partner notification was low among STI cases in the current study, as only 44% delivered the notification slip to their partner notifying them about an STI diagnosis. Among the more than half (56%) who did not deliver the notification slip to their partner, their reasons for failing to tell included a fear of losing their partner and embarrassment. These have been documented as social reasons for failure to notify a partner of an STI in other studies [22,23,31].

## 5. Limitations

The results obtained in this study should not be generalised to all university students. The STI infections were self-reported, and there is a possibility that STIs and risky sexual behaviours were under-reported. Although we limited the recall period to twelve months, participants might have had difficulties in recalling some events that happened in the past, and by doing so may have introduced recall bias. However, during data collection, we assured the students about the anonymity of the study and findings.

## 6. Conclusions 

This study has described the knowledge, self-reported STIs, and preventive practices of university students in South Africa towards STIs. We found that STI knowledge is low in both the male and female students, and prevention practices towards STIs are unacceptable despite the high risky sexual behaviours that are strongly associated with acquiring an STI among the students. The findings indicate that students are vulnerable to STIs due to the low levels of knowledge and risk perceptions about STIs. The male students had a low knowledge level versus the females, while the female students had low risk perceptions compared to the males. Low knowledge about STIs highlights a need for universities to take action and develop education programmes that can create a greater awareness of the risks of STIs among university students. This is of significance given that the study was conducted among health sciences students who are expected to be more knowledgeable and aware of important reproductive health issues. The campus health clinic within universities has an important role to play in planning prevention activities for HIV and other STIs.

This study further reported high intentions among the female students compared to their male counterparts, to notify partners about an STI as well as preferences for partner notification. Patient-initiated partner notification was the most preferred method of partner notification among the students. It is important that health care providers should put in place interventions so that young people can safely inform their partners about STIs. Furthermore, health providers in South Africa should consider strengthening the provider-initiated partner notification methods to provide multiple options that are acceptable to young people. This could improve the success of partner notification for the screening and early diagnosis of STIs to control the spread of STIs in this population.

## Figures and Tables

**Figure 1 ijerph-18-05660-f001:**
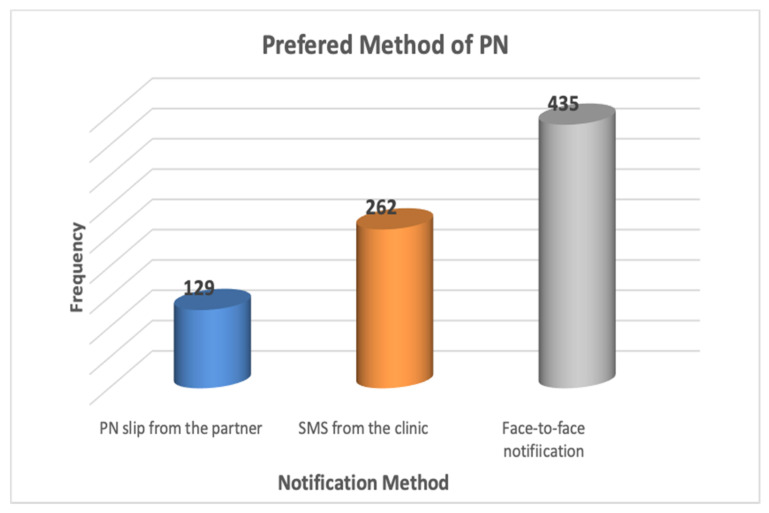
Preferred method of STI notification.

**Table 1 ijerph-18-05660-t001:** Distribution of sociodemographic and behavioural characteristics of university students (*n* = 918).

Characteristic		Frequency (%)
Sex (*n* = 915)	Female	643 (70.3)
	Male	272 (29.7)
	17–20	442 (48.6)
	21–25	391 (42.9)
	>25	77 (8.5)
Field of study (*n* = 911)	Bachelor of Pharmacy	270 (29.6)
	Bachelor Medicine	161 (17.8)
	Bachelor Occupational Therapy	111 (12.2)
	Bachelor of Science	78 (8.6)
	Bachelor Nursing Science	69 (7.6)
	Bachelor Diagnostic Radiography	61 (6.7)
	Other	162 (17.8)
Year of study (*n* = 901)	First	227 (25.2)
	Second	296 (32.8)
	Third	222 (24.6)
	Fourth	135 (15)
	Fifth to sixth	22 (2.4)
Place of residence (*n* = 889)	Campus	455 (49.7)
	Off-campus	430 (46.9)
	Other	31 (3.4)
Sexually active	Yes	459 (50.5)
	No	450 (49.5)
Relationship status (*n* = 661)	Steady partner	479 (72.5)
	Casual partner	160 (24.2)
	Married	22 (3.3)
Duration of the relationship (*n* = 540)	1–11 months	189 (35)
	1–3 years	235 (43.5)
	3–5 years	77 (14.3)
	5–10 years	39 (7.2)
Number of sexual partners in the previous 12 months (*n* = 583)	1	421 (71.3)
	2	97 (16.1)
	>2	75 (12.6)
Concurrent sexual partners (*n* = 257)	No	413 (68.3)
	Yes	192 (31.7)

**Table 2 ijerph-18-05660-t002:** Risky sexual behavioural characteristics and condom use in university students by sex.

	Female		Male		*p* Value
	Yes *n* (%)	No *n* (%)	Yes *n* (%)	No *n* (%)	
**Sexual relationships**					
Sexually active	301 (47.3)	335 (52.7)	146 (54.1)	124 (45.9)	0.063
Had more than one sexual partner	77 (14.1)	470 (85.9)	114 (46.0)	134 (54.0)	<0.001 *
Transactional sex in past 12 months	6 (1.0)	565 (99.0)	13 (5.3)	234 (99.7)	<0.001 *
Had one-night stand in past 6 months	22 (4.0)	533 (96.0)	68 (28.1)	174 (71.9)	<0.001 *
**Condom use**					
Used condom the last sexual act	248 (55.9)	196 (44.1)	166 (69.7)	72 (30.3)	<0.001 *
Male condoms easily available on campus	531 (92.0)	46 (8.0)	245 (90.7)	25 (9.3)	0.529
Female condoms easily available on campus	133 (23.3)	437 (76.7)	46 (20.4)	180 (79.6)	0.364
Could purchase condoms without feeling embarrassed	358 (63.7)	204 (36.3)	216 (80.6)	52 (19.4)	<0.001 *
Could get condoms from a public place without feeling embarrassed	272 (48.1)	294 (51.9)	175 (65.8)	91 (34.2)	<0.001 *
Always carry a condom	85 (16.3)	436 (83.7)	104 (40.9)	150 (59.1)	<0.001 *
Ever used a female condom	18 (3.2)	551 (72.3)	10 (4.5)	211 (95.5)	0.353
Feel confident to suggest condom use with new partner	478 (93.7)	32 (6.3)	237 (92.6)	19 (7.4)	0.548
**HIV testing and STI**					
Has been tested for HIV in the last 12 months	368 (59.8)	247 (40.2)	126 (47.0)	142 (53.0)	<0.001 *
Know partner’s HIV status	315 (60.7)	204 (39.3)	119 (48.4)	127 (51.6)	0.001 *
Had an STI in the last 12 months	27 (4.5)	573 (95.5)	13 (5.6)	239 (94.8)	0.678

* significant at *p* < 0.05.

**Table 3 ijerph-18-05660-t003:** Reported preventive sexual behaviours and perceptions of risk of acquiring HIV.

		Female	Male	*p*-Value
**Chances of refusing sex if condom is not used**				0.000 *
Likely Unlikely		425 (79.7)	158 (61.2)	
		108 (20.3)	100 (38.8)	
**Chances of discussing HIV testing with partner**				0.001 *
Likely		459 (78.5)	178 (67.2)	
Unlikely		125 (21.5)	87 (32.8)	
**Chances of asking partner to go for HIV test**				
Likely		439 (76.8)	155 (59.4)	
Unlikely		133 (23.3)	106 (40.6)	
**Perceived risk of being infected with HIV**				0.012 *
Very worried		304 (49.2)	144 (53.5)	
Worried		113 (18.3)	63 (23.4)	
Not worried		201 (32.5)	62 (23.1)	
**Perceived risk of contracting STIs**				0.265
Likely		78 (12.9)	44 (16.8)	
Unlikely		525 (87.1)	218 (83.2)	
**How important is it to the tell partner about STI infection?**				0.004 *
Not important		12 (2.1)	12 (2.1)	
Important		60 (30.3)	37 (15.7)	
Very important		510 (87.6)	187 (79.2)	

* significant at *p* < 0.05.

**Table 4 ijerph-18-05660-t004:** Students’ knowledge of STI symptoms by sex.

Item	Female	Male	*p*-Value
	Yes *n* (%)	Yes *n* (%)	
Itching in genital area	317 (49.3)	121 (44.5)	0.183
Discharge	329 (51.2)	129 (47.4)	0.301
Pain during urination	309 (48.1)	138 (50.7)	0.459
Genital ulcers or open sores	290 (45.1)	104 (38.2)	0.055
Pain during intercourse	229 (35.6)	76 (27.9)	0.024 *
A person can have an STI without symptoms	421 (70.6)	198 (77.0)	0.054

* significant at *p* < 0.05.

**Table 5 ijerph-18-05660-t005:** Partner notification practices among students who self-reported STI in the past 12 months.

	Frequency	Percent
**Ever diagnosed with STIs**		
No	815	95.3
Yes	40	4.7
**STIs symptoms experienced**		
Itching in genital area	9	34.6
Discharge	7	26.9
Pain when urinating	6	23.1
Genital ulcers or open sores	2	7.69
Pain during intercourse	2	7.69
**Informed sex partner of STI**		
No	14	37.84
Yes	23	62.16
**Received PN slip during consultation for STI symptoms**		
No	12	66.67
Yes	24	33.33
**Delivered PN slip to partner**		
No	14	56
Yes	11	44
**Reasons for failing to tell**		
I would be embarrassed	6	37.5
Fear of losing partner	4	25.00
Partner would refuse to have sex	2	12.50
Partner would blame me	1	6.25
Could not locate partner	1	6.25
**Informed of an STI by partner**		
No	18	72.00
Yes	7	28.00
**Received PN slip from partner with STI**		
No	13	52.00
Yes	12	48.00

**Table 6 ijerph-18-05660-t006:** Perceptions of and intentions to use STI partner notification.

Statement	Female			Male			*p*-Value	OR(95%CI)
	No *n* (%)	Not Sure *n* (%)	Yes *n* (%)	No *n* (%)	Not Sure *n* (%)	Yes *n* (%)		
If you have an STI, could you tell your partner about the infection?	12 (2.1)	60 (10.3)	510 (87.6)	12 (5.1)	37 (15.7)	187 (79.2)	0.55 0.95 *	Ref Not sure 0.13 (0.75–1.17) Yes 1.38 (0.95–2.01)
If have an STI, would you deliver a PN slip to your partner?	20 (3.4)	-	573 (96.6)	18 (7.4)	-	226 (92.6)	0.014 *	Ref Yes 0.43 (0.23–0.84)
If have an STI, would you deliver a PN slip to your ex-sexual partner?	157 (26.8)	174 (29.7)	255 (43.5)	55 (22.6)	67 (27.6)	121 (49.8)	0.55 0.10	Ref Not sure 0.14 (0.57–1.72) Yes 1.38 (0.95–2.01)
If your partner delivers a PN slip that request you for STI treatment, would you find that easy?	80 (13.4)	115 (19.5)	395 (67.0)	38 (15.9)	47 (19.7)	154 (64.4)	0.54 0.34	Ref Not sure 0.85 (0.51–1.42) Yes 0.81 (0.53–1.24)
How easy would it be deliver a PN slip to your partner?	251 (42.8)	101 (17.2)	234 (39.9)	124 (51.2)	42 (17.4)	76 (31.4)	0.42 0.02	Ref Not sure 0.84 (0.55–1.3) Yes 0.66 (0.47–0.92)
Would you prefer an SMS from a doctor sent to your partner to get STI treatment?	184 (31.7)	67 (11.5)	330 (56.8)	56 (23.3)	26 (10.8)	158 (65.8)	0.19 0.02 *	No Ref Not sure 1.4 (0.85–2.23) Yes 1.5 (0.23–0.41)
Do you think an SMS would work better to notify partners to get STI treatment?	202 (34.4)	85(14.5)	300 (51.1)	62 (25.8)	36 (15.0)	142 (59.2)	0.14 0.03 *	No Ref Not sure 0.86 (0.70–1.05) Yes 0.59 (0.36–0.96)

* significant at *p* < 0.05.

## Data Availability

A dataset will be submitted upon request by the Editor.

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
