# Peer review of "A Comparative Analysis of Risky Sexual Behaviors, Self-Reported Sexually Transmitted Infections, Knowledge of Symptoms and Partner Notification Practices among Male and Female University Students in Pretoria, South Africa"

_ijerph, 2021, doi:10.3390/ijerph18115660_

Round 1
Reviewer 1 Report
The manuscript is written quite well, explanatory, with good ideas and results. Nevertheless, I have specific mandatory recommendations to improve its overall quality.
- All the keywords used should be MeSH descriptors. Please verify and correct them.
- Line 40: The word "burden" has been repeated from the previous phrase. Please, use a synonym.
- Lines 88-89: I cannot understand what the authors are trying to say here. Please clarify and rephrase.
- Lines 103-105: “however, most of the literature does not address knowledge and preventive practice of STIs other than HIV”.
- Gender differences are highlighted in the Title but not in the Abstract and not even in the Introduction and Conclusions. I suggest to add some considerations about this investigated construct within these paragraphs.
- Please make an accurate check for plagiarism, since there are some short sentences that turn out to be plagiarized.
- The authors indicated that the questionnaire used was “standardized” and “developed by referring to previous tools and constructs obtained from the review of the literature”. Nevertheless, no data or remarks have been provided on a possible validation of it.
- In the paragraph Methods, the authors did not sufficiently clarify which were the main investigated constructs and how many items they used to measure each construct.
- I recommend adding the complete version of the questionnaire in Supplementary materials.
Author Response
|
Reviewer 1 comments |
Authors response |
|
All the keyword used should be MesSH descriptors. Please verify and correct them. |
MeSH thesaurus was used to correct the keywords in line 31 and 32 |
|
Line 40: The word ”burden” has been repeated from the previous phrase. Please use a synonym. |
The repetition of burden has been removed and the new sentence reads as “In South Africa, the burden of disease due to STIs is one of the largest in the world, with an estimated 23,175 million new cases of syphilis, 4.2 million cases of chlamydia, and 6.2 million cases of gonorrhoea reported in 2017” |
|
Line 88-89: I cannot understand what the authors are trying to say here. Please clarify and rephrase. |
The sentences in line 88-89 were rephrased and reads as: “Research has noted that patient-initiated partner notification interventions have had barriers to its uptake, because index cases under reported the number of sexual partners, due to fear of moral judgement [29, 30].” |
|
Lines 103-105: “however, most of the literature does not address knowledge and preventative practice of STIs other than HIV” |
Line 103-105 has been corrected and reads as:” Furthermore, knowledge, attitude and preventive practice of students on HIV has been extensively studied [31]; however, most of the literature does not address knowledge and preventive practice of STIs.” |
|
Gender differences are highlighted in the Title but not in the Abstract and not even in the Introduction and Conclusions. I suggest to add some considerations about this investigated construct within these paragraphs. |
Highlighted areas where the gender differences that were already in the content and the new additions in the content. Line 323-326; 328; 343-343; 351-353; 362-363; 413; 416-418; |
|
Please make an accurate check for plagiarism, since there are some short sentences that turn out to be plagiarized. |
The manuscript has been submitted to Turnitin software to rule out plagiarism. The grammatical errors were reviewed by the scientific editor |
|
The authors indicated that the questionnaire used was “standardized” and “developed by referring to previous tools and constructs obtained from the review of the literature”. Nevertheless, no data or remarks have been provided on the possible validation of it. |
The study did not consider validating tools that were already validate, especially because they were used in an English speaking context within a health science University population. |
|
In the paragraph Methods, the authors did not sufficiently clarify which were the main investigated constructs and how many items they used to measure each construct. |
The data collection sections is updated and the constructs and items were included in line 134-154 |
|
I recommend adding the complete version of questionnaire in Supplementary materials. |
The questionnaire is attached |
|
|
|
Reviewer 2 Report
The manuscript that is presented for evaluation tries to evaluate risk behaviors, self-reported STI, knowledge and practices of notification to couples among university students. Therefore, it is a relevant issue at the public health level that requires research to advance this knowledge.
I make two observations. It is indicated that a bivariate logistic regression analysis will be performed to measure association forces, but the dependent variables are not defined. Also, no results from this regression analysis are shown.
On the other hand, since the title refers to gender differences, it was to be expected that in addition to showing the results segregated by sex, a theoretical framework on gender would be proposed and the results would be discussed taking into account this perspective, and not only not differences by sex.
Author Response
|
Reviewer 2 comments |
Authors response |
|
I make two observations. It is indicated that a bivariate logistic regression analysis will be performed to measure association forces, but the dependent variables are not defined. Also, no results from this regression analysis are shown. |
Bivariate analysis was conducted and amended in in Table 6. And the narrative of the analysis was written in line 280 to 290. |
|
On the other hand, since the title refers to gender differences, it was to be expected that in addition to showing the results segregated by sex, a theoretical framework on gender would be proposed and the results would be discussed taking into account this perspective , and not only not differences by sex. |
This study did not employ gender frameworks during conceptualization hence the analysis and background did not use a framework. The researchers followed survey methodologies of other health research surveys that are in the field of public health. |
|
|
|
Round 2
Reviewer 2 Report
After the second review of the article I present the following suggestions:
If this study does not employ gender theoretical frameworks, the title should be modified so as not to confuse readers, specifying that the study simply analyzes differences by sex.
In the method section, it remains unspecified which are the independent and dependent variables in the logistic regression analysis. When calculating the Odds ratios and their confidence intervals, it is not necessary to calculate the p-values ​​between the analyzed variables. In addition, since the results of the Odds Ratios and their 95% confidence intervals have been incorporated in Table 6, the p values ​​would already be repetitive, so they could be suppressed.
When writing the results, do not talk about gender, but about sex. When presenting the results of the logistic regression, indicate whether the risks / probabilities are significant or not, taking into account the confidence intervals, not the p-values.
In the discussion many results are repeated again. There is no discussion about the results of the model.
Author Response
The Editor in Chief
MDPI - International Journal of Environmental Research and Public Health
RE: CORRECTIONS OF COMMENTS MADE BY REFEREES
Please receive the Manuscript with the change title: “A comparative analysis of risky sexual behaviours, self-reported sexually transmitted infections, knowledge of symptoms and partner notification practices among university students in Pretoria, South Africa”
We address the issues raised below:
Gender differences was removed from the title and replaced with “A comparative analysis”
Sex (male/female) is the dependent variable since the analysis is focusing the differences by sex
We corrected the narrative of the regression analysis and removed the p-values.
We did not use the model or framework hence there is no discussion on the model. The aspects of the results that is minimum and is used as a background to comparing the findings with existing literature
Thank you in anticipation for considering this manuscript.
Yours Sincerely
Professor Mathildah Mpata Mokgatle
Department of Public Health
18 May 2021